# Successful Hemispherotomy in a Patient with Encephalopathy with Continuous Spikes and Waves during Sleep Related to Neonatal Thalamic Hemorrhage: A Case Report with Intracranial Electroencephalogram Findings

**DOI:** 10.3390/brainsci11070827

**Published:** 2021-06-22

**Authors:** Shimpei Baba, Tohru Okanishi, Toshiki Nozaki, Naoki Ichikawa, Kazuki Sakakura, Mitsuyo Nishimura, Takahiro Yonekawa, Hideo Enoki, Ayataka Fujimoto

**Affiliations:** 1Department of Child Neurology, Comprehensive Epilepsy Center, Seirei Hamamatsu General Hospital, Hamamatsu 430-8558, Japan; okanishipediatrics@gmail.com (T.O.); enokih.neuropediatr@gmail.com (H.E.); 2Epilepsy and Surgery, Comprehensive Epilepsy Center, Seirei Hamamatsu General Hospital, Hamamatsu 430-8558, Japan; tonod03sm069@gmail.com (T.N.); swfff464@gmail.com (N.I.); ksakakura1126@yahoo.co.jp (K.S.); ataka_fuji@sis.seirei.or.jp (A.F.); 3Division of Clinical Laboratory, Seirei Hamamatsu General Hospital, Hamamatsu 430-8558, Japan; zmittuu@gmail.com; 4Department of Pediatrics, Mie University Graduate School of Medicine, Tsu 514-8507, Japan; tyonekawa@clin.medic.mie-u.ac.jp

**Keywords:** epileptic encephalopathy, encephalopathy with continuous spikes and waves during sleep (ECSWS), neonatal thalamic hemorrhage, intracranial electroencephalogram, hemispherotomy

## Abstract

Neonatal thalamic hemorrhage is a strong risk factor for developing encephalopathy with continuous spikes and waves during sleep (ECSWS), even when not accompanied by widespread cortical destruction. The efficacy and indication of resective epilepsy surgery in such patients has not yet been reported. A 4-year-old boy was diagnosed with ECSWS based on strong epileptiform activation during sleep and neurocognitive deterioration. He had a history of left thalamic hemorrhage related to a straight sinus thrombosis during the newborn period. He presented with daily absence seizures that were refractory to medical treatment. At age 5, he underwent intracranial electroencephalogram (EEG) recording using depth and subdural strip electrodes placed in the left thalamus and over bilateral cortex, respectively. Interictal and ictal epileptiform discharges were observed in the thalamus, always preceded by discharges in the left or right parietal lobe. Left hemispherotomy successfully normalized the EEG of his unaffected hemisphere and extinguished his seizures. This is the first case report documenting resective epilepsy surgery in a patient with ECSWS due to neonatal thalamic injury without widespread cerebral destruction. Based on intracranial EEG findings, his injured thalamus did not directly generate the EEG abnormalities or absence seizures on its own. Patients with ipsilateral neonatal thalamic injury and even mild lateralized cortical changes may be candidates for resective or disconnective surgery for ECSWS.

## 1. Introduction

Encephalopathy with continuous spikes and waves during sleep (ECSWS) is an age-dependent epileptic encephalopathy that develops during childhood [1,2,3]. ECSWS is characterized by an electroencephalogram (EEG) pattern with strong activation of epileptiform activity during non-rapid eye movement sleep [4]. Clinical symptoms are variable; most children develop epileptic seizures and can present with variable deterioration in cognitive, language, behavioral, and/or motor aspects during development [1,2]. The etiology, clinical expression, effective treatment, and prognosis have not been fully clarified [1,2]. Moreover, no consensus has been reached regarding the terminology related to the name of the syndrome, concepts, and diagnostic criteria [5].

Recently, several studies revealed that early thalamic injuries (e.g., neonatal stroke, hypoxic-ischemic encephalopathy, and periventricular hemorrhagic infarction) are strongly associated with the later development of ECSWS [6,7]. Most patients with ECSWS also have widespread cerebral destruction and often exhibit hemi-activation of epileptiform activity during sleep when the lesion is lateralized to either hemisphere. Some authors have reported that patients with ECSWS due to structural brain abnormalities benefit from resective epilepsy surgery, such as hemispherotomy or multi-lobar resection [8,9]. Kersbergen et al. reported that patients who acquired neonatal thalamic hemorrhage with a straight sinus thrombosis are at high risk of developing ECSWS and related epilepsy, even when they do not exhibit widespread cerebral destruction, such as encephalomalacic or porencephalic changes [10]. However, the efficacy of resective epilepsy surgery in such patients has not yet been investigated.

Herein, we present a case of ECSWS that was well-controlled by a hemispherotomy. The patient experienced left thalamic hemorrhage during the neonatal period, secondary to a straight sinus thrombosis, without widespread cerebral destruction. To avoid confusion, in this manuscript, we used ECSWS when referring to the epileptic encephalopathy with neurocognitive deterioration and “continuous spikes and waves during sleep (CSWS)” when referring to the EEG characteristics of ECSWS.

## 2. Case Presentation

The boy was delivered by spontaneous delivery at 37 weeks of gestation, without asphyxia. On the third day after birth, the patient exhibited fever and was transferred to a nearby emergency hospital. The boy was diagnosed with left thalamic hemorrhage with ventricular perforation, which was related to a straight sinus thrombosis (Figure 1A,B). The boy developed a subsequent hydrocephalus at 12 days of age, requiring the placement of a ventriculoperitoneal shunt. Right hemiparesis remained as sequelae, and the boy’s motor development was delayed; his neurocognitive development from infancy to early childhood was age-appropriate.

At 3 years of age, the boy developed epilepsy. His seizure semiology included daily absence seizures and bilateral tonic-clonic seizures that were often prolonged. Carbamazepine, valproate, and levetiracetam were initiated, but had no effect. At the age of 4 years, the boy was referred to our hospital for evaluation and further treatment. The boy could walk alone, but fine motor movements of the right fingers and right foot were disabled. He also exhibited somnolence and difficulty in memorizing, even for a short period, and could not establish conversations with others. Brain magnetic resonance imaging (MRI) revealed cystic changes and hemosiderin deposition in the left thalamus (Figure 1C). Regarding the cortex, atrophy and hyperintense lesions in T2/fluid-attenuated inverted recovery imaging were observed in the white matter of the left hemisphere, as well as atrophy of the left hippocampus (Figure 1D); however, widespread brain destruction (e.g., encephalomalacic or porencephalic changes) was not confirmed. No atrophy was observed on the right hemisphere. During video-EEG monitoring, numerous generalized interictal spikes and wave discharges were observed. The appearance of these discharges increased during sleep, and the spike-and-wave index was visually estimated as approximately 50–85% (Figure 2A). Atypical absences that lasted 3–10 s appeared hourly or within minutes (Figure 2B), often accompanied by a preceding EEG change in the left parieto-occipital region. His developmental quotient, measured using the Kyoto scale of psychological development [11], declined from 80 (at 3 years 9 months) to 63 (at 4 years 7 months). Based on these findings, the boy was diagnosed with ECSWS related to a neonatal thalamic hemorrhage. Lamotrigine, ethosuximide, high-dose benzodiazepine therapy, and oral corticosteroids were administered; however, the improvement in EEG findings was limited, and atypical absences, neurocognitive deterioration, and somnolence persisted.

At 5 years of age, the boy was scheduled to undergo resective or palliative epilepsy surgery. He underwent intracranial EEG recording to assess whether his seizures were generated in his thalamic lesion or in the unilateral or bilateral hemispheres. We placed subdural strip electrodes in the frontal, parietal, and parieto-temporal lobes of both hemispheres, as well as depth-electrodes in the left thalamus and corpus callosum (Figure 3A–C). We had decided in advance that if seizures arose from the non-thalamus areas, we would proceed to perform resective or pallative epilepsy surgery. If seizures arose from the left thalamus, we would withdraw the intracranial electrodes. Interictal epileptiform discharges were observed most frequently in the left and right parietal lobes and confirmed in multiple areas, such as the frontal and parieto-temporal lobes (Figure 3D,E). Epileptiform discharges were also seen in the left thalamus and corpus callosum; however, the shape of each discharge was visually dull and always appeared in synchronization with those from the left or right parietal lobe. Epileptiform discharges often spread over both hemispheres, the left thalamus, and the corpus callosum, and appeared rhythmically and continuously, which was assumed to be characteristic of CSWS (Figure 3E). Numerous atypical absences were observed. In each seizure, 1.5–2 Hz bilateral synchronous spike-and-wave discharges appeared for 5–15 s. Most of them started from the left parietal lobe and occasionally from the right parietal lobe, but preceding EEG changes were never seen in the left thalamus (Figure 3F,G). From these findings, we concluded that the atypical absences were not generated directly from the left thalamus but presumably from the cerebrum. As we could not discard the possibility that the right hemisphere could generate seizures alone, we chose to perform a total corpus callosotomy. After the surgery, his intellectual deterioration was ameliorated; however, atypical absences and somnolence persisted. Moreover, monthly focal tonic seizures of the right upper limb, accompanied by right head-turning, newly appeared. Video-EEG monitoring was again recorded, and we confirmed that the interictal epileptiform discharges and ictal EEG changes in every absence seizure were lateralized to the left hemisphere (Figure 4A,B). At 6 years of age, the boy underwent a left peri-insular hemispherotomy (Figure 5A). The histopathological findings of the resected hippocampus and left temporal lobe were compatible with hippocampal sclerosis and focal cortical dysplasia type IIId associated with perinatal bleeding, respectively. Immediately after the hemispherotomy, all pre-existing seizures disappeared, and the interictal EEG of the right hemisphere was normalized (Figure 5B). Fine motor skills of the right fingers remained deteriorated, but the gross motor function of the right limbs and speech function were maintained. Six months after the hemispherotomy, we discontinued several antiepileptic drugs without seizure recurrence or ECSWS exacerbation; his neurocognitive function gradually improved.

## 3. Discussion

We described the case of a boy with ECSWS who underwent a successful hemispherotomy. To the best of our knowledge, this is the first report documenting resective epilepsy surgery in a patient whose epilepsy was assumed to be due to a neonatal thalamic injury. Several studies have documented the efficacy of resective surgery in patients with early thalamic injury, but the patients also had widespread severe hemispheric destruction [8,9]. Indeed, our patient had lateralized cortical- and subcortical damage on the ipsilateral cortex of the injured thalamus, but the extent of the injury was comparatively milder than those of previously reported cases [8,9]. We believe that our experience will be of help to clinicians who consider resective surgery for patients with ECSWS related to early thalamic injury without widespread cortical destruction.

The role of the thalamus in generating and maintaining generalized epileptic seizures/epileptiform discharges has been recognized in studies using EEG-functional MRI [12], volumetric MRI [13], and positron emission tomography [14]. However, controversy remains regarding whether the trigger of the epileptic seizures lies in the thalamus or cortex. For example, Dalic et al. recently analyzed EEG findings in patients with Lennox-Gastaut syndrome (LGS) with simultaneous recordings from depth electrodes placed in the thalamic centromedian nucleus (CM) and scalp electrodes and found that most of the onsets of generalized fast activities were observed earlier in the scalp electrodes, whereas the onset of slow spikes and waves were variable [15]. Velasco et al. analyzed the EEG onset of various types of seizures observed in patients with LGS using the depth electrodes placed in the CM and scalp electrodes and reported that the onset of generalized tonic, tonic-clonic, and atypical absence seizures occurred simultaneously in the CM and cortex; however, the EEG onset of myoclonic seizures was observed earlier in the CM [16]. Martin-Lopez et al. studied the ictal EEG (by depth-EEG in the CM and scalp EEG) in two patients with idiopathic generalized epilepsy and one patient with frontal lobe epilepsy. They found that the onset of the epileptic discharges preceding absence seizures was not constant between the two patients with idiopathic generalized epilepsy; leading epileptic discharges occurred earlier in the cortex in one patient, whereas earlier in the thalamus in the other. They also speculated that the thalamus is capable of autonomous epileptogenesis, as the thalamus possesses epileptiform discharges of its own [17]. We found that EEG abnormalities and the absence seizures in our patient arose from the cortex, always preceding the abnormal thalamic activity, and speculated that it was unlikely that the injured thalamus itself generated these abnormalities. We believe our experience will add new insights into the relationship between the thalamus and cortex in the pathogenesis of ECSWS and absence seizures.

It is noteworthy that the epileptiform discharges and spike-and-wave discharges preceding the absence seizures were observed not only in the left hemisphere, which was ipsilateral to the injured thalamus but also in the contralateral right hemisphere. Moreover, the interictal epileptiform discharges and ictal EEG changes were lateralized to the left hemisphere on scalp-electrode EEG recordings after the corpus callosotomy (Figure 4). As the relationship between the thalamus and cortex is regarded as ipsilateral [18], it can be speculated that the corpus callosum played an important role in our patient, presumably by propagating the abnormal electrical activity from the left thalamus and left cortex to the right hemisphere. Hence, in patients with ECSWS due to unilateral thalamic injury, with/without a corresponding cortical injury, a corpus callosotomy could be a good tentative treatment, as the unaffected hemisphere might be protected from the abnormal electrical activity generated in the damaged thalamus/cortex. We also note that our patient required a hemispherotomy for complete seizure control and improvement in alertness; the propagation of abnormal electrical activities via other pathways (e.g., the anterior commissure) might disrupt the function on the unaffected hemisphere. While the etiology is a neonatal thalamic injury, lateralized hemispheric epilepsy network existed beyond the lesion [19,20,21]. Therefore, surgery would be considered to be one of the treatment options in case of lateralized EEG findings in addition to MR changes.

Our report has some limitations. First, it might be inappropriate to apply the interpretation of our data to the pathogenesis of ECSWS without structural etiology or absence seizures in patients with so-called “idiopathic” generalized epilepsy, as our patient had an apparent thalamic injury and might have different mechanisms in the development of epilepsy. Second, as the follow-up period after the hemispherotomy was short, we could not confirm improvement in the long-term prognosis; however, we did not observe an exacerbation of seizure/EEG control despite the rapid discontinuation of several antiepileptic drugs. Third, our data suggested that the lateralized structural MR abnormality was sufficient to generate epilepsy and that the thalamus was not the major issue when considering resective or disconnective surgery; intracranial EEG recording might not be mandatory for patients with ECSWS and similar brain lesion as our patient.

In conclusion, we presented the first case of a patient with ECSWS related to a neonatal thalamic hemorrhage who underwent hemispherotomy. Our data suggest that patients with ipsilateral neonatal thalamic injury and even mild lateralized cortical changes may be candidates for resective or disconnective surgery for ECSWS. We hope our data provide new insights toward clarifying the mechanism of ECSWS/absence seizures.

## Figures and Tables

**Figure 1 brainsci-11-00827-f001:**
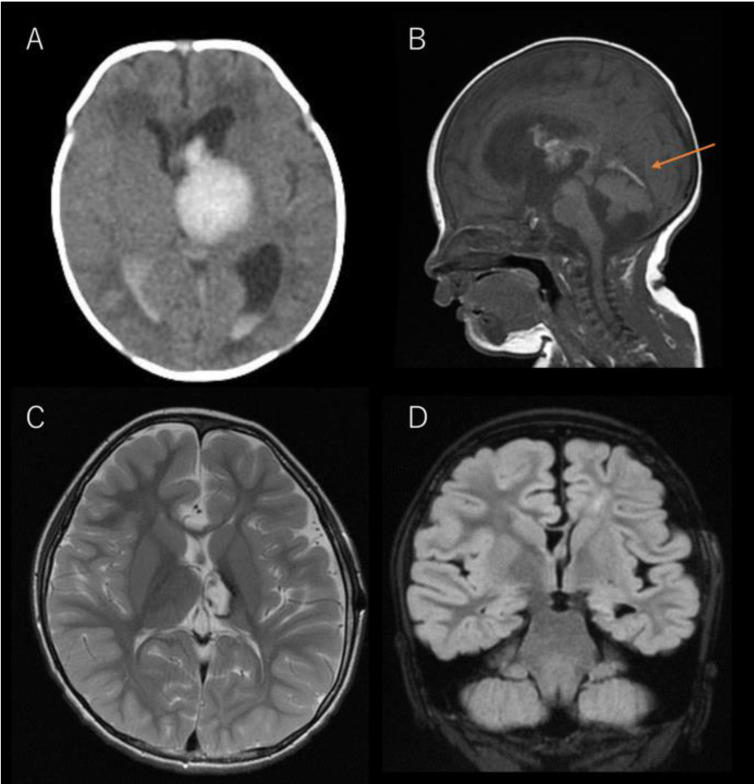
Brain magnetic resonance imaging (MRI) and computed tomography (CT) of the patient. (**A**) Axial CT image at 5 days of age. Thalamic hemorrhage perforating into the lateral and third ventricles is observed. (**B**) Sagittal T1-weighted image at 29 days of age. A high-intense, linear lesion is observed over the tentorium cerebelli (arrow), which is presumed to be the straight sinus thrombosis. (**C**) Axial T2-weighted image at 4 years of age. Cystic changes and hemosiderin deposition in the left thalamus are confirmed. No atrophy is observed on the right hemisphere. (**D**) Coronal fluid-attenuated inverted recovery image at 4 years of age. Slight atrophy of the left hemisphere, diffuse high-intensity lesions of the left cerebral white matter, and hippocampal atrophy are observed.

**Figure 2 brainsci-11-00827-f002:**
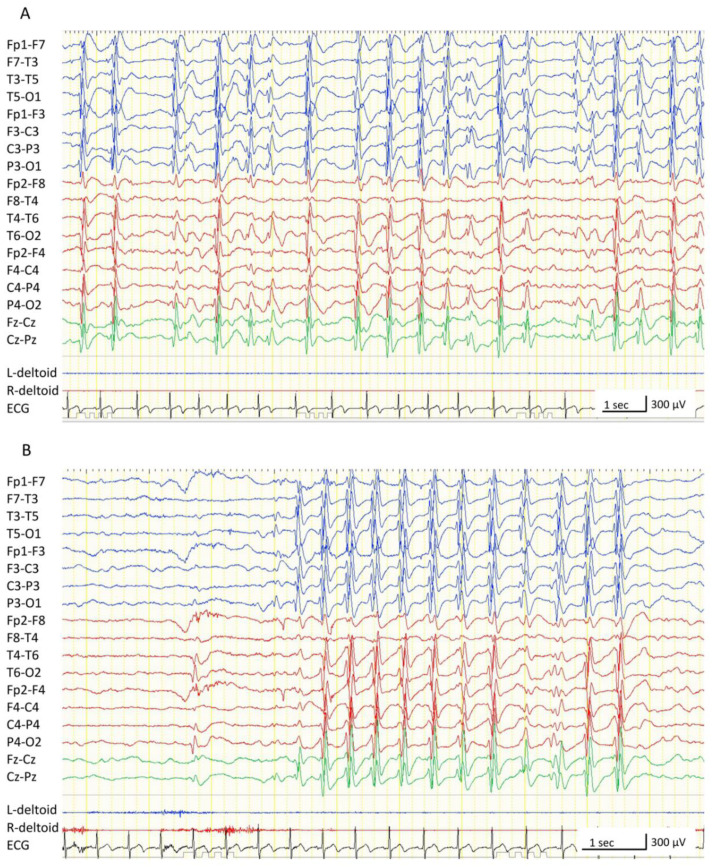
Scalp electroencephalogram (EEG) at 4 years of age. All EEGs were recorded with the following settings: low-cut filter, 1.6 Hz; high-cut filter, 60 Hz. (**A**) Interictal sleep EEG. Continuous epileptiform discharges are observed predominantly in the left hemisphere. (**B**) Ictal EEG of an absence seizure. 2.5- to 3 Hz spike and wave discharges in the left parieto-occipital region precede the generalized spike and wave discharges compatible with ictal EEG characteristics of absence seizures.

**Figure 3 brainsci-11-00827-f003:**
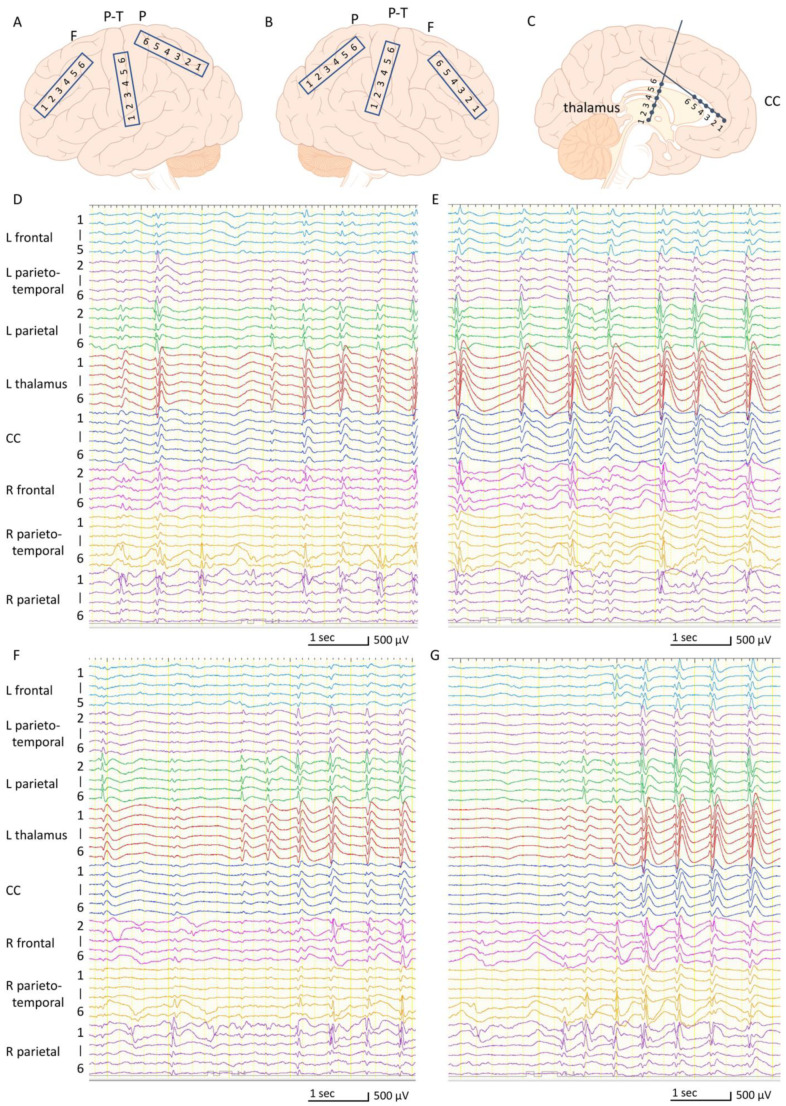
Intracranial electroencephalograms (EEGs) at 5 years of age. All EEGs were recorded with the following settings: low-cut filter, 1.6 Hz; high-cut filter, 60 Hz. All EEGs are displayed in a referential montage, with the referential electrode placed on the forehead. EEGs recorded by strip electrode No. 6 placed in the left frontal lobe, No. 1 in the left parieto-temporal lobe, No. 1 in the left parietal lobe, and No. 1 in the right frontal lobe were removed due to the poor quality of the recording. (**A**–**C**) Schemes indicating the placement of the subdural strip and depth electrodes. (**D**) Interictal sleep EEG. Epileptiform discharges are observed predominantly in the electrodes placed over the left parietal, right parietal, and right parieto-temporal lobes. Epileptiform discharges are also found in the left thalamus, but are preceded by those from the left or right parietal lobe. (**E**) Interictal sleep EEG. Epileptiform discharges are widespread over both hemispheres, the left thalamus, and the corpus callosum. (**F**,**G**) Ictal EEG of an absence seizure. Most ictal EEG changes started from the left parietal lobe (**F**) and often from the right parietal lobe as well (**G**). Abbreviations: CC, corpus callosum; F, frontal lobe; P, parietal lobe; P-T, parieto-temporal lobe.

**Figure 4 brainsci-11-00827-f004:**
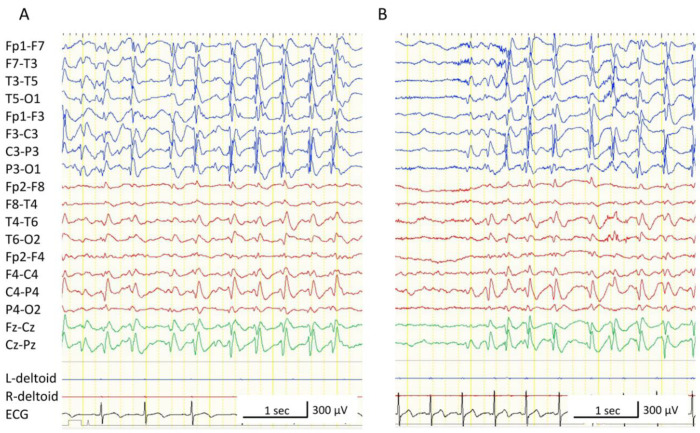
Scalp electroencephalograms (EEG) after the corpus callosotomy. All EEGs were recorded with the following settings: low-cut filter, 1.6 Hz; high-cut filter, 60 Hz. (**A**) Interictal EEG. Epileptiform discharges are lateralized to the left hemisphere. (**B**) Ictal EEG of an absence seizure. Ictal changes are observed in the left hemisphere but not in the right hemisphere.

**Figure 5 brainsci-11-00827-f005:**
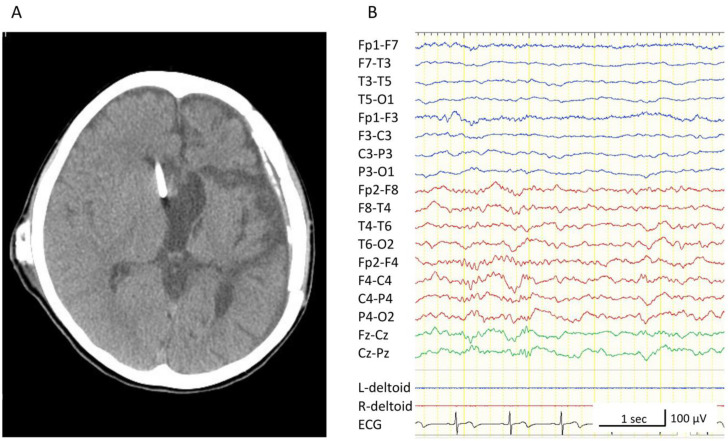
Brain computed tomography (CT) and scalp electroencephalogram (EEG) after the left hemispherotomy. The EEG was recorded with the following settings: low-cut filter, 1.6 Hz; high-cut filter, 60 Hz. (**A**) Brain CT after the left hemispherotomy. (**B**) Interictal EEG. The appearance of epileptiform discharges is decreased; the formation of sleep spindles is confirmed in the right hemisphere.

## Data Availability

The data presented in this report are available from the corresponding author on reasonable request.

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
