# Peer review of "Successful Hemispherotomy in a Patient with Encephalopathy with Continuous Spikes and Waves during Sleep Related to Neonatal Thalamic Hemorrhage: A Case Report with Intracranial Electroencephalogram Findings"

_brainsci, 2021, doi:10.3390/brainsci11070827_

Round 1

Reviewer 1 Report

Case Report Review

Hemispherotomy in a patient with CSWS related to neonatal thalamic hemorrhage: case report with intracranial EEG findings.

Case presentation

Would emphasize that clear moderate lateralized atrophy could be seen in the left hemisphere with the other changes suggesting unilateral cortical and subcortical damage

No atrophy was seen on the right

Would remove widespread brain damage was not confirmed

Would remove he was scheduled to undergo respective or palliative surgery

Why was intracranial monitoring completed?  Would be useful to indicate

Appears to be a per-insular hemipherotomy. This could be mentioned

Discussion

While the etiology is a neonatal thalamic injury, lateralized hemispheric changes do exist which appear mild to moderate in degree

Surgery would be considered with lateralized EEG findings in addition to MR changes

In limitations need to suggest that the lateralized structural MR abnormality was sufficient to generate his epilepsy and that the thalamus was not the major issue as confirmed by intracranial eeg

Intracranial eeg was it necessary?

Conclusion

Patients with Ipsilateral Neonatal thalamic injury and even mild lateralized cortical changes may be candidates for resective or disconnective surgery

Author Response

Response to reviewer #1

We greatly appreciate the reviewers’ comments and suggestions, which have helped us improve the quality of our manuscript. The manuscript has been revised to address the issues raised by the reviewers. The changes made and our point-by-point responses to the comments from the reviewer are summarized below. Our changes to the text are shown in red font in the revised manuscript.

Comment #1:

> Would emphasize that clear moderate lateralized atrophy could be seen in the left hemisphere with the other changes suggesting unilateral cortical and subcortical damage

> No atrophy was seen on the right

> Would remove widespread brain damage was not confirmed

Response #1:

Thank you very much for your comments. As the reviewer suggested, lateralized atrophy could be identified in the left hemisphere, whereas no atrophic changes were found in the right. The phrase “widespread brain damage was not confirmed” was inappropriate. We apologize for the lack of clarity.

However, we would like to indicate that current evidence regarding resective- or disconnective epilepsy surgery for ECSWS with cortical lesion is solely based on the experiences of patients with extensive brain malformation or destruction, e.g., lateralized encephalomalacia or porencephaly (Peltora ME, et al. Epilepsia 2011 (reference No. 8 in our manuscript), Marashly A, et al. Front Neurol 2020 (reference No. 9)). Hence, the extent of brain injury is relatively milder in our patient than those in the previously reported cases. Few patients had lateralized thalamic injury with ipsilateral cortical atrophy like our patient, but they were treated with corpus callosotomy (Peltora ME, et al. Epilepsia 2011).

Considering the above, we did not remove the sentence “widespread brain damage (e.g., encephalomalacic or porencephalic changes) was not confirmed” on Page 3, Lines 96-97 to emphasize the difference between our case and the previously reported cases. Instead, we revised the phrase “widespread cerebral (or cortical) lesions” to “widespread cerebral (or cortical) destruction” on Page 1, line 20-21, Page 1, Line 31, Page 2, line 58,  and Page 8, Line 221, and “widespread brain damage” to “widespread brain destruction” in Page 3, Lines 96-97.

Furthermore, we have added the following sentences on Page 8, Lines 217-219, to emphasize that our patient had lateralized cerebral damage, which was different from those of previously reported cases in its extent.

“Indeed, our patient had lateralized cortical- and subcortical damage on the ipsilateral cortex of the injured thalamus, but the extent of the injury was comparatively milder than those of previously reported cases [8,9].”

Finally, we have added “No atrophy was observed on the right hemisphere” on Page 3, Lines 97-98 and to the legend of figure 1 accordingly.

Comment #2:

> Would remove he was scheduled to undergo respective or palliative surgery

> Why was intracranial monitoring completed?  Would be useful to indicate

Response #2:

Thank you very much for your comments. We apologize for this omission.

We have added the following sentence:

“We had decided in advance that if seizures arose from the non-thalamus areas, we would proceed to perform resective or pallative epilepsy surgery. If seizures arose from the left thalamus, we would withdraw the intracranial electrodes”. (pages 4-5, lines 121-123)

Comment #3:

> Appears to be a per-insular hemipherotomy. This could be mentioned

Response #3:

This is also an important point. We apologize for this omission.

We have added this term. (page 5, lines 143-144)

Comment #4:
> While the etiology is a neonatal thalamic injury, lateralized hemispheric changes do exist which appear mild to moderate in degree
> Surgery would be considered with lateralized EEG findings in addition to MR changes

Response #4:

Thank you very much for pointing this out. As reviewer#1 mentioned, “lateralized EEG findings in addition to MR change”, sometimes, we experienced complicated epilepsy networks with a localized MRI lesion.

Therefore, we have added the following sentences:

 “While the etiology is a neonatal thalamic injury, lateralized hemispheric epilepsy network existed beyond the lesion [19-21]. Therefore, surgery would be considered as one of the treatment options in case of lateralized EEG findings in addition to MR changes”. (page 9, lines 261-264)

   Furthermore, we have added the following articles in our reference:

  1. Scholly, J.; Staack, A.M.; Kahane, P.; Scavarda, D.; Régis, J.; Hirsch, E.; Bartolomei, F. Hypothalamic hamartoma: Epilepto-genesis beyond the lesion? Epilepsia 2017, 58, 32-40.
  2. Besseling, R.M.; Jansen, J.F.; de Louw, A.J.; Vlooswijk, M.C.; Hoeberigs, M.C.; Aldenkamp, A.P.; Backes, W.H.; Hofman, P.A. Abnormal Profiles of Local Functional Connectivity Proximal to Focal Cortical Dysplasias. PLoS One 2016, 11, e0166022.
  3. Sakakura, K.; Fujimoto, A.; Arai, Y.; Ichikawa, N.; Sato, K.; Baba, S.; Inenaga, C.; Matsumura, A.; Ishikawa, E.; Enoki, H.; Okanishi, T. Posttraumatic epilepsy may be a state in which underlying epileptogenicity involves focal cortical dysplasia. Epilepsy Behav 2021, 114, 107352.

Comment #5:

> In limitations need to suggest that the lateralized structural MR abnormality was sufficient to generate his epilepsy and that the thalamus was not the major issue as confirmed by intracranial eeg

> Intracranial eeg was it necessary?

Response #5:

   Thank you for your comments. We completely agree that the lateralized structural MR abnormality was sufficient to generate epilepsy, and the thalamic injury was not the major issue when considering the resective surgery.

Regarding the necessity of the intracranial EEG, we needed to confirm whether the epileptiform discharges and the epileptic seizures arose from the cortex or the thalamus, as there remains a controversy regarding whether the thalamus can generate epileptiform discharges and/or epileptic seizures (Please check the second paragraph of the discussion section). Based on our experience, we consider that intracranial EEG recording may not be mandatory for patients with ECSWS related to thalamic- and ipsilateral cortical injury. We hope our experience is useful to clinicians who care for patients with ECSWS related to similar brain damage as our patient.

   We have added the following sentences on Page 9, Lines 272-275:

“Third, our data suggested that the lateralized structural MR abnormality was sufficient to generate epilepsy and that the thalamus was not the major issue when considering resective or disconnective surgery; intracranial EEG recording might not be mandatory for patients with ECSWS and similar brain lesion as our patient.”

Comment #6:

> Patients with Ipsilateral Neonatal thalamic injury and even mild lateralized cortical changes may be candidates for resective or disconnective surgery

Response #6:

  Thank you for your suggestion. We have removed “Neonatal thalamic injury without widespread cerebral lesions might not be a contraindication for resective epilepsy surgery for ECSWS” in abstract and “… that neonatal thalamic injury without widespread cortical lesions is not a contraindication for resective epilepsy surgery” in conclusion; instead, we have added the following sentence according to your suggestion.

“…patients with ipsilateral neonatal thalamic injury and even mild lateralized cortical changes may be candidates for resective or disconnective surgery for ECSWS.” (page 9, lines 277-279)

Reviewer 2 Report

The authors present a first patient with continuous SW during sleep due to remote symptomatic thalamic hemorhage and slight malformation of cortical developments and hippocampal sclerosis on the left treated by complete callosotomy and functional left hemispherotomy. I have read this paper with great interest and I could only congratulate the authors for the well done job. 

Author Response

Response to reviewer #2

Thank you for reviewing our manuscript and for your kind comments. We are encouraged to keep working on daily medical practices.
